# NKG2D Fine-Tunes the Local Inflammatory Response in Colorectal Cancer

**DOI:** 10.3390/cancers15061792

**Published:** 2023-03-16

**Authors:** Sophie Curio, Wanzun Lin, Christian Bromley, Jenny McGovern, Chiara Triulzi, Gustav Jonsson, Ghita Ghislat, Santiago Zelenay, Nadia Guerra

**Affiliations:** 1Department of Life Sciences, Imperial College London, London SW7 2BX, UK; 2The University of Queensland Frazer Institute, Faculty of Medicine, The University of Queensland, Woolloongabba, QLD 4102, Australia; 3Cancer Inflammation and Immunity Group, Cancer Research UK Manchester Institute, The University of Manchester, Alderley Park, Manchester M20 4BX, UK

**Keywords:** NKG2D, tumor immunology, inflammation, colorectal cancer, immunotherapy

## Abstract

**Simple Summary:**

NKG2D is a type of immune receptor that is expressed on several subsets of lymphocytes and involved in the recognition and elimination of infected cells and cancer cells. The discovery of NKG2D as a therapeutic target for cancer has led to the development of novel immunotherapy strategies that aim to activate the immune system to fight various types of cancer. Our study explores the role played by NKG2D in colorectal cancer. We show that high expression of NKG2D is associated with decreased survival in a subset of patients, in contrast with the protective role observed in other cancer types. Our work highlights that the presence of NKG2D-expressing cells is not always a good prognostic marker in advanced tumors, and that more research is needed to fully understand its mechanisms of action and investigate the efficacy and safety of NKG2D-based therapy in colorectal cancer.

**Abstract:**

Treating colorectal cancer (CRC) is a major challenge due to the heterogeneous immunological, clinical and pathological landscapes. Immunotherapy has so far only proven effective in a very limited subgroup of CRC patients. To better define the immune landscape, we examined the immune gene expression profile in various subsets of CRC patients and used a mouse model of intestinal tumors to dissect immune functions. We found that the NK cell receptor, natural-killer group 2 member D (NKG2D, encoded by *KLRK1*) and NKG2D ligand gene expression is elevated in the most immunogenic subset of CRC patients. High level of *KLRK1* positively correlated with the mRNA expression of *IFNG* and associated with a poor survival of CRC patients. We further show that NKG2D deficiency in the *Apc^min/+^* mouse model of intestinal tumorigenesis led to reduced intratumoral IFNγ production, reduced tumorigenesis and enhanced survival, suggesting that the high levels of IFNγ observed in the tumors of CRC patients may be a consequence of NKG2D engagement. The mechanisms governing the contribution of NKG2D to CRC progression highlighted in this study will fuel discussions about (i) the benefit of targeting NKG2D in CRC patients and (ii) the need to define the predictive value of NKG2D and NKG2D ligand expression across tumor types.

## 1. Introduction

Colorectal cancer (CRC) is the third most frequent and the second most lethal cancer in the world [1]. Although mortality rate in CRC has declined in the last decade, relapse occurrence is high, and survival is extremely poor at advanced stages [2]. Adjuvant chemotherapy after surgery has remained the main treatment modality during the last two decades. CRC is classified into different consensus molecular subtypes (CMSs) with distinguishing molecular features; CMS1 emerges as the most immunogenic subtype with high lymphocyte infiltration and increased microsatellite instability [3]. CMS1 patients are consequently the most responsive to immune checkpoint inhibition (ICI) among all CRC subsets. While survival rates in CMS1 patients have improved upon combination with immune checkpoint inhibitors, only a fraction of these patients responds to the treatment [4]. Besides ICI, chimeric antigen receptor-T (CAR-T) cell safety and efficacy are also being evaluated in CRC in phase I clinical trials [5,6,7,8]. One alternative to targeting tumor antigens in cell therapy relies on the detection of stress-induced ligands, such as ligands for the activating receptor NKG2D, known to be expressed in most cancer types [9]. NKG2D ligands belong to the family of major histocompatibility complex (MHC) class I-related chain A and B (MICA/B) and the UL16-binding proteins (ULBP1–ULBP6) in humans [9]. The extensive polymorphism of the *MICA/B* genes together with the diversity of stress inducers upregulating MICA/B and the 6 ULBPs, has challenged our understanding of their regulation, especially in the tumor microenvironment [10,11]. NKG2D ligands can stimulate adaptive and innate immune responses by engaging NKG2D on αβ T cells, γδ T cells, iNKT cells, and innate lymphoid cells (ILCs), including natural killer (NK) cells. Targeting NKG2D in CAR cell therapies employing NK cells, γδ T cells or αβ T cells has proved to be a viable approach by several preclinical studies [12,13,14] and is currently in its early clinical phase (NCT04623944, NCT05247957), including in patients with refractory metastatic CRC (NCT05213195, NCT05248048) [15,16,17,18,19]. While encouraging anti-tumor activity was recorded against hematological malignancies [20], limited data are available in the case of solid tumor and metastatic CRC in particular [21].

The intra-tumor and inter-individual heterogeneity of NKG2D and NKG2D ligand expression in cancer make it particularly difficult to predict the safety and efficacy of the NKG2D-based platforms. Despite its well-established anti-tumor function, the NKG2D axis is being carefully considered in immunotherapy as the presence of NKG2D ligands on healthy tissue subject to chronic inflammation raises concerns about possible on-target off-tumor toxicity [22,23,24]. NKG2D-expressing cells are implicated in the progression of autoimmune diseases and inflammatory diseases, including inflammatory bowel disease [25]. Further, recent studies have shown that the expression of certain NKG2D ligands is associated with poor survival in CRC patients [26]. Our previous findings in a model of inflammation-driven cancer highlighted the risks of chronic liver tissue damage and subsequent tumor progression when NKG2D ligand expression is sustained [27,28]. In line with this, high expression levels of certain human NKG2D ligands were associated with poor prognosis in advanced-stage hepatocellular carcinoma [29]. In the gut tissue, we recently showed that NKG2D-mediated immune activation contributes to tumor progression [30]. Specifically, in the *Apc^min/+^* model of intestinal tumorigenesis, we demonstrated a role for NKG2D in IL-17A-producing γδT cells, a particular subset defined as CD27^-^Vγ6 T cells with pro-tumor function. Here, we investigated further the role of NKG2D in CRC and determined the gene expression profile of NKG2D and most ligands according to the cancer molecular subtypes. We show that NKG2D ligand expression shapes the inflammatory landscape and disease prognosis of CRC by enhancing IFNγ secretion in the tumor microenvironment (TME).

## 2. Materials and Methods

### 2.1. Mice

As previously described [30], NKG2D-heterozygous (*Klrk1^+/−^*) mice were crossed with *Apc^min/+^* mice to generate *Klrk1^+/+^;Apc^min/+^, Klrk1^−/−^;Apc^min/+^, Klrk1^+/+^;Apc^+/+^* and *Klrk1^−/−^;Apc^+/+^* mice and to study the impact of NKG2D-deficiency in the intestine. The health status of *Apc^min/+^* mice was checked and evaluated frequently. Disease severity was assessed using a scoring scheme that included parameters such as appearance, natural behavior, provoked behavior, body condition, and tumor score. Mice were humanely euthanized when they had reached the experimental endpoint. Mice were bred and maintained in the animal facility at Imperial College London in a specific pathogen-free environment. Work was carried out in compliance with the British Home Office Animals Scientific Procedures Act 1986 and the EU Directive 2010 and sanctioned by Local Ethical Review Process (PPL 70/8606).

### 2.2. Lymphocyte Isolation from Intestinal and Tumor Tissue

To isolate lymphocytes from tumor tissue, the small intestine was harvested and separated in duodenum, jejunum, and ileum. Data presented in this study relate to ileal tumors. The small intestine was dissected longitudinally, and tumors were counted and carefully excised. Tumor tissue was digested in digestion buffer (RPMI-1640 containing 5% FCS, 25 mM HEPES, 150 U/mL collagenase IV (Sigma Aldrich, St. Louis, MO, USA), and 50 U/mL DNase (Roche, Basel, Switzerland) and incubated on an incubator shaker for 37 °C at 200 rpm for 30 min. Dissociated tissue was filtered through a 100 µm strainer. To inhibit enzyme activity, PBS containing 10% FCS and 5mM EDTA was added. Cells were pelleted by centrifugation (10 min at 500× *g*) before proceeding to flow cytometry staining.

Control intraepithelial lymphocytes and lamina propria lymphocytes were collected from healthy wild-type mice. After harvesting the small intestine, the remaining fat and intestinal content were removed. The intestine was flushed with PBS and opened longitudinally. Mucus was removed and tissue was cut into small pieces and transferred to mucus removal buffer (10 mL of HBSS containing 1 mM DTT, 2% FCS, 100 U/mL penicillin and 100 µg/mL streptomycin). Tubes were vortexed for 30 min and the solution was replaced. Tissue was then incubated for 20 min on an incubator shaker (37 °C, 200 rpm). After this, tissue was transferred to a dissociation buffer (10 mL of HBSS containing 1 mM EDTA, 2% FCS, 100 U/mL penicillin, and 100 µg/mL streptomycin) and incubated for 5 min (37 °C, 200 rpm). The solution was filtered through a 100 µm strainer, and the tissue washed twice with fresh dissociation solution, resulting in the removal of all epithelial cells. To isolate lamina propria lymphocytes, the remaining tissue was cut into small pieces and digested in 7 mL of digestion buffer (100 U/mL collagenase VIII (Sigma Aldrich) and 50 U/mL DNase (Roche)) for 45 min (37 °C, 100 rpm). The digested tissue was taken up into 10 mL stripettes and filtered through a 100 µm cell strainer. Both solutions were centrifuged for 10 min at 500× *g*.

### 2.3. Flow Cytometry 

To detect intracellular proteins, cells were stimulated with PMA (final concentration 600 ng/mL), ionomycin (final concentration 100 ng/mL) and Brefeldin A (final concentration 10 µg/mL) in RPMI-1640 containing 5% FCS, 100 U/mL penicillin and 100 µg/mL streptomycin prior to antibody staining. Cells were incubated with anti-mouse CD16+CD32 (BD, Franklin Lakes, NJ, USA) for 20 min, and LIVE/DEAD Fixable Aqua Dead Cell Stain kit (ThermoFisher, Waltham, MA, USA) or the Zombie NIR Fixable Viability Dye (BioLegend, San Diego, CA, USA) for 20 min prior to adding the antibody mix to prevent unspecific binding by the Fc receptor and detect dead cells. The antibody mix was then added and incubated for 30 min at 4 °C. When cytokines or transcription factors were stained, cells were permeabilized using the eBioscience Transcription Factor Fixation/Permeabilization kit (ThermoFisher, Waltham, MA, USA) prior to staining. Antibodies to detect intracellular antigens were added and incubated for 30 min. Samples were acquired on a BD LSR Fortessa and analyzed using FlowJo software. Antibodies used in this study are listed in Appendix A.

### 2.4. Analysis of TCGA Gene Expression Database

#### 2.4.1. Data Acquisition and Processing

The log2-normalised gene expression data (file: COADREAD.uncv2.mRNAseq_RSEM_normalized_log2.txt) and corresponding clinical information for The Cancer Genome Atlas CRC patients were downloaded from the Broad Institute Firehose portal (https://gdac.broadinstitute.org, accessed on 13 January 2021). Totally, 480 tumor samples and 41 adjacent healthy tissues from patients that did not receive treatment prior to sample collection were included. A Transcripts Per Million (TPM) method was used to normalize TCGA HTseq-counts. This project was performed in accordance with the guidelines provided by TCGA. The CMSclassifier R package was used to stratify tumor samples into CMS subtypes 1–4. The expression of NKG2D and its ligands were compared between normal and tumor tissues using the ggplot2 package in R software (version 4.2.0). The correlation between *KLRK1* and immune checkpoints was analyzed by ggstatsplot package in R software (version 4.2.0). The correlation coefficient (r) and *p*-value were calculated by Pearson correlation coefficient. For survival analyses, we used the survival R (version 3.2.7) package and stratified patients according to individual gene expression levels or signature scores using median cut-offs.

#### 2.4.2. Gene Set Enrichment Analysis (GSEA)

GSEA was performed to reveal the significantly altered signaling pathways between high- and low-*KLRK1* groups, which was identified in the MSigDB Collection (h.all.v7.5.symbols.gmt, downloaded on 20 July 2022). Gene set permutations were set at 1000 repeats.

### 2.5. Statistical Analysis

Statistical analysis was performed using Python (Pandas) or GraphPad Prism. Shapiro–Wilk tests were performed to test for normality. Significance was determined using Mann–Whitey U or unpaired t-tests unless stated otherwise. Data visualization was performed using GraphPad Prism (version 8.4.2).

## 3. Results

### 3.1. NKG2D and NKG2D Ligand Expression Is Upregulated in Colorectal Cancer Patients

We determined the expression of NKG2D and NKG2D ligands in intestinal tumorigenesis by comparing their expression pattern in primary tumor and healthy tissue of CRC patients using The Cancer Genome Atlas (TCGA) Colon Adenocarcinoma (COAD) and Colorectal Adenocarcinoma (COADREAD) transcriptomics datasets. Ligands belonging to the ULBP family (*ULBP1-3*) were drastically upregulated in most patients (Figure 1a). The expression of *MICA* and *MICB*, members of the MHC class I polypeptide-related sequence (MIC) family, was higher in normal samples compared with other ligands and increased in most patients (Appendix A). The increased expression of NKG2D ligands in the tumor tissue indicates that NKG2D-expressing cells can be selectively activated and are likely to contribute to the immune responses taking place within the tumor microenvironment (TME).

CRC is a heterogeneous disease, and the response to immunotherapy in CRC patients is highly dependent on the subtype of cancer. According to the CMS classification [3], tumors with high infiltration of immune cells are classified as CMS1 and are typically associated with response to ICI. To understand how NKG2D ligand expression differs between CRC subtypes and whether NKG2D ligand expression can potentially be used as a biomarker, we compared the expression of *MICA/B* and *ULBP-1/2/3* in CMS1-4. All ligands were expressed at significantly higher levels in CMS1 patients compared with the other subtypes (Figure 1b). Similarly, *KLRK1*, which encodes NKG2D, was differentially expressed in different subtypes of CRC, being significantly increased in patients classified as CMS1 compared to CMS2, 3 and 4 (Figure 1c). These data suggest that the NKG2D/NKG2D ligand activation pathway may contribute to the anti-tumor immune responses in the local inflammation observed in the CMS1 subtype of CRC patients.

### 3.2. NKG2D Expression Associates with an IFNγ Signature

Next, we asked whether *KLRK1* expression could be used as a predictor for survival and found that high expression of *KLRK1* was associated with decreased overall survival in patients with CMS1 (Figure 2a) but not the other CMS subtypes (Appendix A). Specifically, *KLRK1* expression was strongly associated with a IFNγ and NK/ILC1 signature defined by the expression of *CD16, GZMB, ITGAE* and *SLC4A10* or *IL12B, IFNG* and *TBX21*, respectively (Figure 2b). The decreased survival of patients expressing high levels of *KLRK1* (Figure 2a), together with the strong correlation between *KLRK1* expression and type 1 inflammatory response (Figure 2b,c and Appendix A), suggests that this inflammatory response may not be beneficial. Indeed, high expression of genes associated with an IFNγ signature was associated with a modestly decreased survival (Figure 2c). The expression of PD-1 (encoded by *PDCD1*), PD-L1 (encoded by *CD274*) (Figure 2d) and other immune inhibitory checkpoints (CTLA4, LAG3, PD-L2 and TIGIT) (Appendix A) were strongly correlated with *KLRK1* expression. Together, these data suggest that NKG2D contributes to the persistent stimulation of intratumoral type 1 immunity and is associated with poor survival in CRC patients displaying high *KLRK1* expression.

### 3.3. Tumorigenesis in the Apc^min/+^ Mouse Correlates with Increased NKG2D Expression and Type 1 Immunity

To understand the mechanism linking high NKG2D expression with poor prognosis in CRC, we studied the function of NKG2D-expressing tumor-infiltrating cells in a mouse model of intestinal cancer—the *Apc^min/+^* mouse model of spontaneous multiple intestinal adenomas and colon carcinoma [31,32]. To understand the development of the disease, we analyzed the tumor-infiltrating lymphocytes (TILs) at two timepoints. The early disease timepoint corresponds to adenomas visible in 18 to 20 weeks old mice when they start to develop symptoms, including anemia and weight loss. The typical late disease endpoint corresponds to mice that reached 25 weeks of age and above. We found that between 18–20 weeks and the disease endpoint, the frequencies of group 1 ILCs and CD8^+^ T cells increased from 12 to 20% (Figure 3a,b). Additionally, the frequency of myeloid cells (CD3^−^CD19^−^NK1.1^−^CD11b^+^) increased, and B cells decreased, while the frequencies of CD4^+^ T cells and γδT cells were comparable (Figure 3a). We next asked whether NKG2D was upregulated in the populations that gain prominence at endpoints, ILCs and CD8^+^ T cells. Due to the difficulty of separating healthy cells from tumors in the tumor-bearing intestine, we compared NKG2D expression on healthy intraepithelial lymphocytes (IEL) and lamina propria leukocytes (LPL) isolated from healthy *Apc^+/+^* mice with TILs isolated from *Apc^min/+^* mice. We found that CD8^+^ T cells and group 1 ILCs significantly upregulated NKG2D and accumulated in the tumors of 18–20-week-old mice (Figure 3c). We next studied the functionality of these cells over time and measured the production of the pro-inflammatory cytokine IFNγ. We found that CD8^+^ T cells, but not group 1 ILCs, produced significantly higher levels of IFNγ in advanced tumors at the disease endpoint than at earlier 18–20 week time point (Figure 3d and Appendix A). Next, we measured the expression of PD-1 on these CD8^+^ T cells and found that, in line with the increased IFNγ production, PD-1 was significantly upregulated on CD8^+^ T cells isolated from mice at disease endpoint compared with 18–20 weeks (Figure 3e). Together these data indicate that CD8^+^ T cells expand and/or accumulate at late stages of disease during which they are activated through NKG2D and produce high levels of IFNγ. Chronic engagement of NKG2D and IFNγ production may consequently contribute to PD-1 upregulation and immune cell exhaustion. 

### 3.4. NKG2D Deficiency Reduces IFNγ Production in the Tumor Microenvironment

To probe the requirement for NKG2D in TIL activation in this model, we used NKG2D-deficient mice on the *Apc^min^* background. We previously reported that *Apc^min/+^* mice deficient for *Klrk1* (*Apc^min/^;Klrk1^−/−^*) survive longer than *Apc^min/+^;Klrk1^+/+^* mice and displayed a lower tumor burden [30]. When comparing the frequencies of IFNγ-producing CD8^+^ T cells in *Apc^min/+^;Klrk1^+/+^* and *Apc^min/+^;Klrk1^−/−^* mice, we found that the frequency of CD8^+^ T cells was significantly decreased in the absence of NKG2D (Figure 4a,b) demonstrating the importance of NKG2D in CD8^+^ T cells enrichment. We divided CD8^+^ T cells into three categories based on their ability to produce IFNγ into three categories: ‘high’ (>20% positive cells), ‘medium’ (10–20% positive cells) and ‘low’ (<10% positive cells) (Figure 4c,d). When comparing *Apc^min/+^;Klrk1^+/+^* with *Apc^min/^;Klrk1^−/−^*, we observed a higher fraction of tumor-infiltrating IFNγ-high producing CD8^+^ T cells in *Apc^min/+^;Klrk1^+/+^* than *Apc^min/+^;Klrk1^−/−^* mice (Figure 4c,d). Mice falling into the ‘low’ category were evenly distributed among genotypes (7 *Apc^min/+^;Klrk1^+/+^* and 7 *Apc^min/+^;Klrk1^−/−^*). Conversely, only 4 *Apc^min/+^;Klrk1^+/+^* mice were classified as ‘medium’ versus 10 *Apc^min/+^;Klrk1^−/−^* mice. The higher proportion of IFNγ^+^ CD8^+^ T cells in NKG2D-sufficient versus -deficient mice implies an accumulation of IFNγ in the TME in an NKG2D-dependent manner. To determine whether CD8^+^ T cells correspond to chronically activated cells, we measured PD-1 expression and showed that it was slightly lower in NKG2D-deficient mice compared to *Apc^min/+^;Klrk1^+/+^* controls (Figure 4e). Along with the higher proportion of IFNγ^+^ CD8^+^ T cells in NKG2D-sufficient versus -deficient mice, these data confirm that the higher levels of IFNγ in the TME are a consequence of NKG2D engagement.

## 4. Discussion

In this study, we demonstrate that high expression of the NKG2D receptor enhances the production of IFNγ in a mouse model of intestinal cancer and that *KLRK1* expression in primary CRC patients strongly correlates with the expression of genes associated with an IFNγ response. Notably, we show that both NKG2D and its ligands are upregulated specifically in the more immunogenic CMS1 CRC tumors, and that expression of these genes is associated with the upregulation of immune checkpoint molecules, including PD-1. Gene expression analysis revealed that high expression of *KLRK1* is associated with reduced survival in CMS1 patients which led us to postulate that whilst increased expression of NKG2D and NKG2D ligands enhances immune-mediated tumor control at early stages of cancer development, they may contribute to immune exhaustion and dysfunction in the TME at advanced stages of CRC, specifically in CMS1 patients.

All ligands for NKG2D were found to be upregulated in CRC patients compared to matched normal tissues and more so in the CMS1 subset. We also found that high expression of the *ULBP-1*, *ULBP2* genes in CRC patients and of *MICA* in *APC*-mutant CRC patients correlated with reduced survival (not shown). These observations are in line with recent analyses of *ULBP1* mRNA expression in 438 patients with colon adenocarcinoma [33] and of *MICA* mRNA and protein expression in 96 CRC patients [26], although in contradiction with other MICA-related conclusions [34,35]. Additionally, we cannot exclude that at the protein level, NKG2D ligands are downregulated from the cell surface and/or shed as soluble molecules leading to a suboptimal cytolytic activity towards tumor cells and contributing to tumor progression. However, a recent study of 24 primary colon cancer patients showed no correlations between the gene expression of all ligands and the level of NKG2D expression on circulating NK and NKT cells [36]. Hence, the prognostic value of NKG2D ligand expression remains to be established by integrating the nature, the expression level, the level of regulation (mRNA, membrane and extracellular proteins), the localization of each ligand and their co-expression in the TME and adjacent tissue. 

The role of IFNγ in tumor progression is complex and most likely time- and dose-dependent. IFNγ is typically an indicator of good prognosis in colorectal cancer [37], yet studies have suggested that low concentrations of IFNγ can negatively impact the anti-tumor response, whereas high levels lead to tumor regression [38]. To understand the mechanism resulting in IFNγ-mediated tumor progression, we used a mouse model of colorectal cancer where we previously showed that NKG2D-deficiency improved survival [30]. We observed that T cells expanded and/or accumulated at late stages of disease in an NKG2D-dependent manner where they produced high levels of IFNγ. We have previously shown that NKG2D ligands are highly expressed in that model and that similarly, NKG2D-expressing γδT cells that secrete IL-17A accumulate in the TME, contributing to a pro-tumorigenic microenvironment that sustains tumor growth [30]. These conclusions are supported by in vitro studies that use colorectal cancer spheroids to show that blocking NKG2D decreases immune cell infiltration [39], suggesting that immune cells infiltrate the TME in an NKG2D-dependent manner.

In addition, the accumulation of IFNγ-producing CD8^+^ T cells in the TME could be the result of overtly activated clones specific for tumor antigens that contribute to feeding IFNγ to the inflammatory milieu but no longer eliminate tumor cells. Indeed, prolonged IFNγ signaling has been shown to result in tumor progression through the upregulation of inhibitory ligands and receptors, immune cell exhaustion and resistance to immunotherapy [40,41,42]. Thus, it is possible that IFNγ production upon NKG2D engagement led to PD-1 upregulation and immune cell exhaustion in the TME, as seen for the human CRC cohort. A non-exclusive scenario is that IFNγ fosters inflammation and remodeling of the TME to a pro-tumorigenic immune environment through the recruitment of immunosuppressive myeloid cells [43]. Indeed, IFNγ has been shown to be important in driving intestinal inflammation [44], which is a major risk factor for the development of CRC [45]. We argue that NKG2D mediated immune activation may cause local tissue damage as it enhances inflammation, as previously shown in other contexts [27,28,30], favoring tumor growth [46]. These two non-exclusive scenarios may rely on different cell types expressing NKG2D and/or occurring at different localization and time points during tumor progression.

In showing that NKG2D expression is correlated with a IFNγ response while at the same time being associated with reduced survival in patients as well as in a mouse model, our studies highlight the existence of antagonistic effects of NKG2D during cancer progression. Further understanding of the mechanisms leading to the dual outcome of NKG2D activation may open novel therapeutic avenues for personalized therapies based on the expression of NKG2D and NKG2D ligands.

## 5. Conclusions

High expression of NKG2D and genes associated with an IFNγ signature are associated with decreased survival in a subset of colorectal cancer patients. Furthermore, mice lacking NKG2D survive longer and have a reduced frequency of IFNγ-producing CD8^+^ T cells. These novel findings support the observation that NKG2D contributes to the tumor-promoting environment in CRC and highlight the need to better understand the mechanisms underlying NKG2D-mediated cancer immunosurveillance in different subsets of CRC to develop targeted therapies.

## Figures and Tables

**Figure 1 cancers-15-01792-f001:**
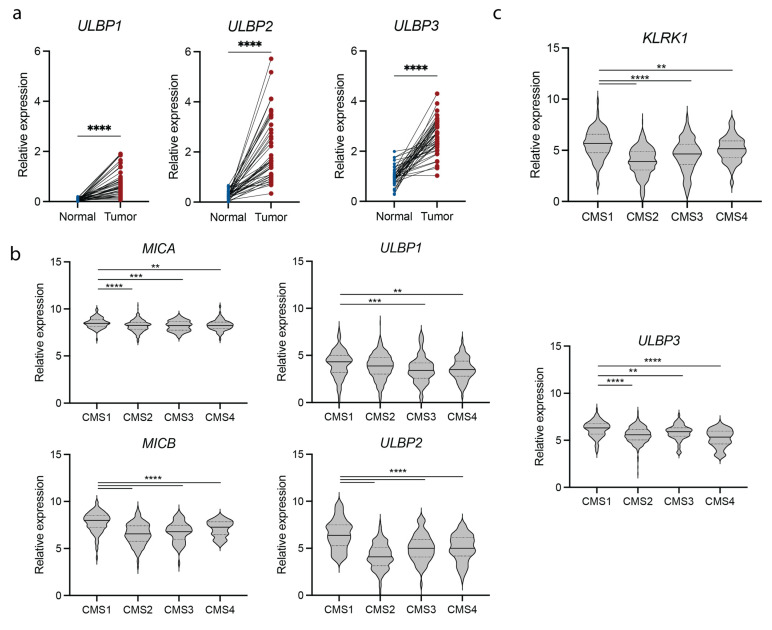
NKG2D and NKG2D ligand expression is increased in CMS1 colorectal cancer patients. (**a**) Relative expression of the NKG2D ligands *ULBP1*, *ULBP2* and *ULBP3* in tumor compared to adjacent healthy tissue of COAD patients. (**b**) Relative expression of NKG2D ligands (*MICA*, *MICB*, *ULBP1*, *ULBP2* and *ULBP3*) in different subtypes of colorectal cancer. (**c**) Relative expression of NKG2D (*KLRK1*) in different subtypes of colorectal cancer. COAD = Colon Adenocarcinoma, CMS = consensus molecular subtype. Statistical significance was determined using paired *t*-test or Wilcoxon matched-pairs signed rank test (**a**) following Shapiro-Wilk normality test or Wilcoxon Rank Sum (**b**,**c**). ** *p* ≤ 0.01, *** *p* ≤ 0.001, **** *p* ≤ 0.0001.

**Figure 2 cancers-15-01792-f002:**
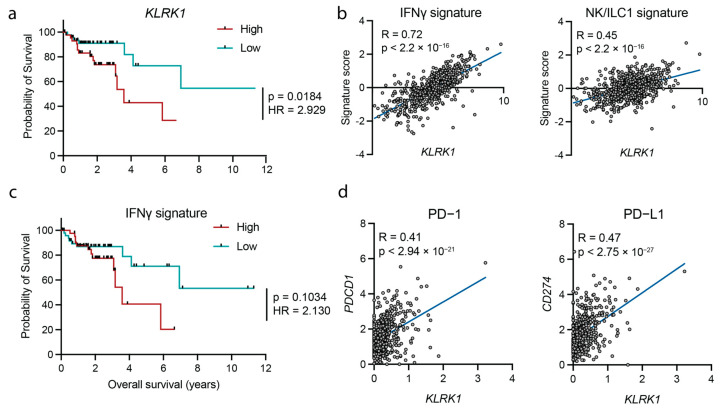
NKG2D expression associates with an IFNγ response and the expression of inhibitory immune checkpoint markers. (**a**) Survival probability in patients expressing high (red) vs. low (teal) levels of *KLRK1*. (**b**) Correlation between an IFNγ signature (left) and an NK/ILC1 signature (right) and *KLRK1* expression. (**c**) Survival probability in patients with a high (red) vs. low (teal) IFNγ signature. (**d**) Correlation between PD-1 (*PDCD1*) (left) and PD-L1 (*CD274*) (right) and *KLRK1* in colorectal cancer patients. COAD = Colon Adenocarcinoma, GSEA = gene set enrichment analysis, HR = hazard ratio. Statistical significance was determined using Log-rank (Mantel-Cox) and Mantel-Haezenszel test comparing upper and lower percentile (**a**,**c**) and linear regression analysis (**b**,**d**).

**Figure 3 cancers-15-01792-f003:**
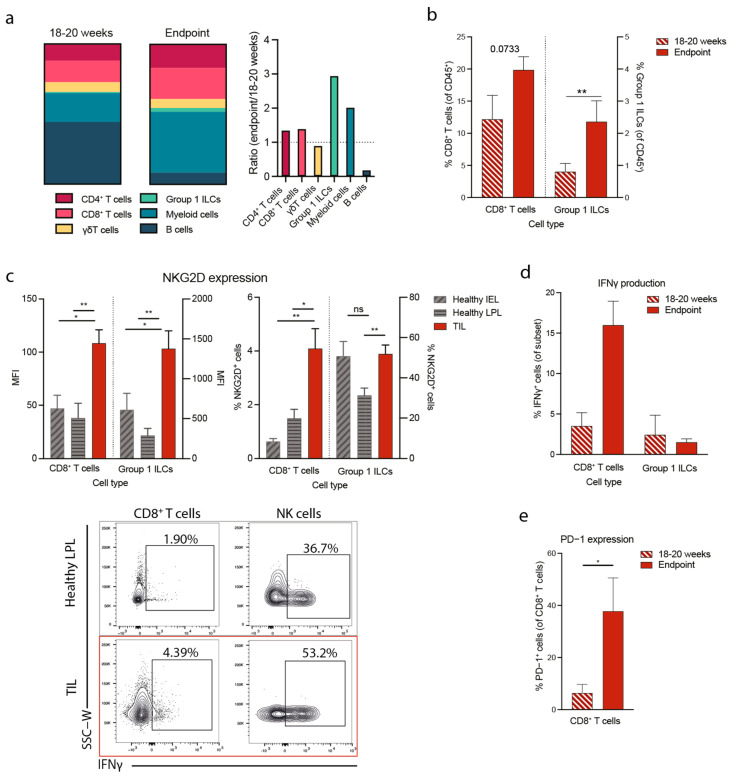
Disease progression in *Apc^min/+^* mice is associated with an increased proportion of IFNγ-producing CD8^+^T cells in the TME. (**a**) Immune composition of *Apc^min/+^* mice at 18–20 weeks (left) and disease endpoint (middle) and ratio between 18–20 weeks and endpoint (right). Cells were gated on Live CD45^+^ lymphocytes with subsequent gating as follows: CD4^+^ T cells: CD3^+^CD4^+^, CD8^+^ T cells: CD3^+^CD8^+^, γδT cells: CD3^+^TCRδ^+^, Myeloid cells: CD3^−^CD19^−^NK1.1^−^CD11b^+^ and B cells: B220^+^CD11b^−^ (**b**) Percentages of CD8^+^ T cells (left) and Group 1 ILCs (right) at 18–20 weeks and disease endpoint. (**c**) MFI of NKG2D (left) and percentage of NKG2D-expressing cells within each subset (right) in healthy IEL, LPL and TIL at 18–20 weeks and representative flow cytometry plots showing IFNγ production by different cell subsets (bottom). (**d**) Percentages of tumor-infiltrating IFNγ-producing CD8^+^ T cells and Group 1 ILCs at 18–20 weeks and disease endpoint. (**e**) Percentages of tumor-infiltrating PD-1-expressing CD8^+^ T cells at 18–20 weeks and disease endpoint. IEL = intraepithelial lymphocytes, LPL = lamina propria lymphocytes, TIL = tumor-infiltrating lymphocytes, ILC = innate lymphoid cell, ns = not significant. Statistical significance was determined using a Mann–Whitney U or unpaired t-test following a Shapiro–Wilk normality test. Bars represent mean ± SEM. * *p* ≤ 0.05, ** *p* ≤ 0.01.

**Figure 4 cancers-15-01792-f004:**
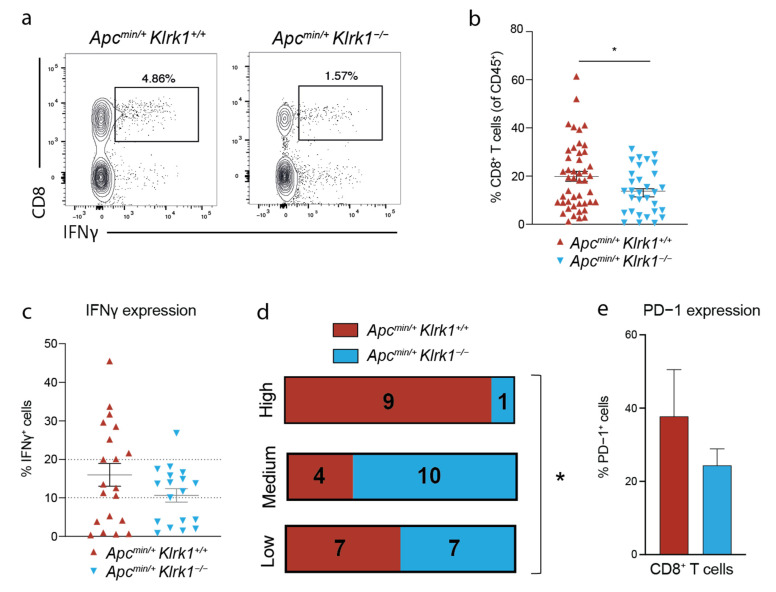
NKG2D deficiency leads to a reduced IFNγ response in *Apc^min/+^* mice. (**a**) Representative flow cytometry plots of tumor-infiltrating IFNγ-producing CD8^+^ T cells in *Apc^min/+^Klrk1^+/+^* and *Apc^min/+^Klrk1*^−/−^ mice at disease endpoint. (**b**) Percentages of tumor-infiltrating CD8^+^ T cells in *Apc^min/+^Klrk1^+/+^* and *Apc^min/+^Klrk1*^−/−^ mice at disease endpoint. (**c**) Percentages of tumor-infiltrating IFNγ^+^ cells within the CD8^+^ T cell subset in *Apc^min/+^Klrk1^+/+^* and *Apc^min/+^Klrk1*^−/−^ mice. (**d**) Categories of IFNγ levels in *Apc^min/+^Klrk1^+/+^* compared to *Apc^min/+^Klrk1*^−/−^ mice. (**e**) Percentages of tumor-infiltrating PD-1 expressing CD8^+^ T cells in *Apc^min/+^Klrk1^+/+^* compared to *Apc^min/+^Klrk1*^−/−^ mice. Statistical significance was determined using Mann Whitney U or unpaired *t*-test following a Shapiro–Wilk normality test (**b**,**c**,**e**) or Chi-square (**d**). Bars represent mean ± SEM. * *p* ≤ 0.05.

## Data Availability

Raw data is available on request.

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
