# Peer review of "NKG2D Fine-Tunes the Local Inflammatory Response in Colorectal Cancer"

_cancers, 2023, doi:10.3390/cancers15061792_

Round 1

Reviewer 1 Report

Overall, the study is well-designed. However, a few considerations seem appropriate:

1) Not having analyzed the cytokine profile of CD4 T lymphocytes, it seems a bit inappropriate to mention a Th1 signature, since the main IFNg-producing cells among those analyzed were CD8, it would be wiser to speak of a TC1-type signature. For the sake of accuracy, we should speak of an IFNg signature.

2) Since it has been suggested that IFNg has a dual role in the immune response to tumors (https://doi.org/10.1186/s40364-020-00228-x), often associated with the dose of IFNg in the tumor microenvironment - IFNg should be quantified to draw some conclusions about the relevance of IFNg in this context.

3) In fig suppl 1 , an extremely curious profile is shown - where IL-10 and IL-15 (among others) appear in tumor tissue associated with high expression of KLRK1; interestingly, it was recently reported that "Interleukin-10 induces interferon-γ-dependent emergence myelopoiesis" (https://doi.org/10.1016/j.celrep.2021 .109887), in this regard, it would be of great interest to measure the expression/production of IL-10 ( associated with the aforementioned expression of IFNg may prove to be extremely important); this observation is reinforced by the presence of anemia and shift to myeloid lineage

4) Authors should show in supp date the gating strategy used for FACS analysis and the negative controls.

Reviewer 2 Report

This is an interresting paper exploriong the role of NKG2D in colorectal carcinogenesis

Comments

Introduction:

1) It could be report that NKG2D blockade prevents immune infiltration and activation in tumor spheroids in a in vitro modelsupporting the dual effect of NKG2D modulation (Courau T et al, Journal for ImmunoTherapy of Cancer, 2019) for introduction of discussion.

Results:

About TCGA data

1) Are the results of KLRK1 expression of normal vs tumour available?

2) Do the author have any data aboput the tumour stage of the patients? if there is difference of distribution of KLRK1 according to stage, that may biaise the survival results

3) Are the tumour samples obtain from primary tumour or metastasis?

4) Is there any details about MSI/dMMR status? Indeed KLRK1 expression is higher in CMS1 subgroup but as for other subgroup there is a wide distribution of KLRK1 expression into the CMS1 subgroup. is there any difference in this sub-group according MSI/dMMR status?

5) Is there any patient enroled in the TCGA analysis treated with immune checkpoint inhibitor? ICI had a dramatic efficacy in MSI/dMMR tumour and completely change the prognosis of the disease in this subgroup.

Chapter 3.2

There is a mistake in the Figure numerotation Line 211. it seems rather Figure 2a than 1d.

Figure 3 legend: ILC is not defined     

Discussion 

It would be interresting to have a larger panel of immune checkpoint expression assessment (not only PD-1) according to NKG2D expression in order to test multiple blockade  

Round 2

Reviewer 2 Report

No supplementary comment